# Blocky Diagonalized Scattering Matrices in Chaotic Scattering with Direct Processes

**Felipe Castañeda-Ramírez** †🆔 **and Moisés Martínez-Mares** *,†🆔

Departamento de Física, Universidad Autónoma Metropolitana-Iztapalapa, Apartado Postal 55-534,
Ciudad de México 09340, Mexico
* Correspondence: moi@xanum.uam.mx
† These authors contributed equally to this work.

**Abstract:** Scattering matrices that can be diagonalized by a rotation through an angle $\theta$ in $2 \times 2$ blocks of independent scattering matrices of rank $N$, are considered. Assuming that the independent scattering matrices are chosen from one of the circular ensembles, or from the Poisson kernel, the $2N \times 2N$ scattering matrix may describe the scattering through chaotic cavities with reduced symmetry in the absence, or presence, of direct processes, respectively. To illustrate the effect of such symmetry, the statistical distribution of the dimensionless conductance through a ballistic chaotic cavity in the presence of direct processes is analyzed for $N = 1$ using analytical calculations. We make a conjecture for $N = 2$ in the absence of direct processes, which is verified by numerical random-matrix theory simulations, and the first two moments are calculated analytically for arbitrary $N$.

**Keywords:** block scattering matrices; chaotic scattering; Poisson kernel; direct processes

## 1. Introduction

Blocky diagonalized scattering matrices has being of interest along the last three decades due to their application in the description of transport properties through wave systems with discrete symmetries. In particular, two-dimensional ballistic cavities with specular (left–right) symmetry are described by diagonalizable scattering matrices by a rotation through an angle $\frac{\pi}{4}$ [1–12]. For those cases, the general structure of the scattering matrix is of the form of a $2 \times 2$ matrix of blocks, each of rank $N$, whose blocks in the diagonal, or in the off-diagonal, are identical. The same structure appears in the scattering matrix associated with elliptically polarized electromagnetic waves scattered by a planar interface between two dielectric media [13]; it becomes diagonal when it is written in the basis of linear polarization. Additionally, we have found that the ensemble of scattering matrices with block symmetry, associated with the set of locally periodic structures of all sizes at the edge of the band, satisfies the same statistics as the one of chaotic cavities with reflection symmetry in the presence of direct processes [14] that give rise to a prompt response in complex scattering [15].

However, there could be chaotic systems in which the diagonalization of the associated random scattering matrix $S$ is obtained from a more general rotation by an angle $\theta$. In the rotated basis, the scattering matrix can be written as

$$S' = R_\theta \, S \, R_\theta^T = \begin{pmatrix} S_1 & 0 \\ 0 & S_2 \end{pmatrix}, \tag{1}$$

where $R_\theta$ is the $2N \times 2N$ rotation matrix given by

$$R_\theta = \begin{pmatrix} \mathbb{I}_N \cos \theta & \mathbb{I}_N \sin \theta \\ -\mathbb{I}_N \sin \theta & \mathbb{I}_N \cos \theta \end{pmatrix}, \tag{2}$$

with $\mathbb{I}_N$ being the identity matrix of dimension $N$. In Equation (1), $S_1$ and $S_2$ are $N \times N$ scattering matrices that we will assume to be statistically independent; they belong to one of the three symmetry classes introduced by Dyson [16]. Due to flux conservation, $S_j$ ($j = 1, 2$) is a unitary matrix ($S_j^\dagger S_j = \mathbb{I}_N$): in the absence of any other restriction, this symmetry is the unitary one, labeled by $\beta = 2$; in addition, $S_j$ is symmetric ($S_j = S_j^T$) in the presence of time reversal invariance. This is the orthogonal symmetry labeled by $\beta = 1$. Finally, $S_j$ is a self-dual matrix of rank $2N$ in the presence of time reversal invariance but has no spin-rotation symmetry; it is labeled by $\beta = 4$.

From Equation (1), the structure of $S$ is of the form

$$S = \begin{pmatrix} r & t \\ t & r' \end{pmatrix}, \tag{3}$$

where

$$r = S_1 \cos^2 \theta + S_2 \sin^2 \theta, \quad r' = S_1 \sin^2 \theta + S_2 \cos^2 \theta, \quad \text{and} \quad t = (S_1 - S_2) \sin \theta \cos \theta. \tag{4}$$

Here, $r$, $r'$, and $t$ are $N \times N$ (or $2N \times 2N$) matrices for $\beta = 1$ and $2$ (or $\beta = 4$). For ballistic cavities connected to two leads, $r$ represents the reflection matrix when incidence is from one side (left) and $r'$, the reflection matrix for incidence from the other side (right); $t$ is the transmission matrix. Although the structure of $S$ in Equation (3) is similar to a one with $\beta = 1$ symmetry, it is not the case. Moreover, the matrix $S$ does not inherit the properties of the constituents matrices $S_1$ and $S_2$. However, $S$ is a unitary matrix because the flux conservation condition should be satisfied; it is also symmetric for the $\beta = 1$ case or self-dual for $\beta = 4$.

According to the optical model of nuclear physics [17,18], the wave amplitudes in complex scattering processes contain two components: a rapid component that arises from the direct processes and a delayed component resulting from the multiple scattering. In chaotic scattering, the first component is described in terms of the ensemble average of the scattering amplitudes, and the second one is a fluctuating part which is studied using random-matrix theory techniques [15]. That is, in the absence of direct processes, $S_j$ belongs to one of the circular ensembles [1,19]: orthogonal (COE) for $\beta = 1$, unitary (CUE) for $\beta = 2$, and symplectic (CSE) for $\beta = 4$. In the presence of prompt responses, the direct processes are quantified by the ensemble average $\langle S_j \rangle$, known as the optical matrix, and the statistical distribution of $S_j$ is given by the Poisson kernel [19].

$$dP_{\langle S_j \rangle}^{(\beta)}(S_j) = \frac{\left[\det\left(\mathbb{I}_N - \langle S_j \rangle \langle S_j \rangle^\dagger\right)\right]^{(\beta N + 2 - \beta)/2}}{\left|\det\left(\mathbb{I}_N - S_j \langle S_j \rangle^\dagger\right)\right|^{\beta N + 2 - \beta}} d\mu_\beta(S_j), \tag{5}$$

where $d\mu_\beta(S_j)$ is the invariant measure, which defines the corresponding circular ensemble for the symmetry $\beta$, which we assume to be normalized to unity (see Appendix A). Note that in the absence of direct processes, $\langle S_j \rangle = 0$, the statistical distribution of $S_j$ is just the invariant measure, which expresses the notion of equal a priori probabilities for $S_j$.

The main purpose of the present paper is to address the statistical properties of systems whose scattering matrix has the structure of Equation (3), in the presence, and hence in the absence, of direct processes. The ensemble average of $S$ is given by

$$\langle S \rangle = \begin{pmatrix} \langle r \rangle & \langle t \rangle \\ \langle t \rangle & \langle r' \rangle \end{pmatrix}, \tag{6}$$

where

$$\langle r \rangle = \langle S_1 \rangle \cos^2 \theta + \langle S_2 \rangle \sin^2 \theta, \quad \langle r' \rangle = \langle S_1 \rangle \sin^2 \theta + \langle S_2 \rangle \cos^2 \theta, \quad \text{and} \quad \langle t \rangle = [\langle S_1 \rangle - \langle S_2 \rangle] \sin \theta \cos \theta. \tag{7}$$

Since we are assuming that $S_1$ and $S_2$ are statistically independent, the statistical distribution of $S$ is given by the product of two independent Poisson kernel distributions; that is,

$$dP_{\langle S \rangle}^{(\beta)}(S) = dP_{\langle S_1 \rangle}^{(\beta)}(S_1) \, dP_{\langle S_2 \rangle}^{(\beta)}(S_2). \tag{8}$$

The consequences of the distribution (8) can be illustrated for the transmission coefficient (dimensionless conductance) $T$, defined by $T = \text{tr}(tt^\dagger)$, which is directly related to the conductance $G$ in electronic devices through the Landauer formula; for each spin polarization [20],

$$G = \frac{e^2}{h} T. \tag{9}$$

For that purpose, we determine the statistical distribution of $T$ by analytical calculations for $N = 1$, which is of great interest and relevance to the experiments [21]. Numerical calculations are used to verify a conjecture for the distribution of $T$ for $N = 2$, in the absence of direct processes. The average and variance of $T$ are also obtained analytically for arbitrary $N$, in the absence of direct processes.

## 2. Results

### 2.1. Statistical Distribution of T in the Presence of Direct Processes for N = 1

For $N = 1$, the scattering matrix of the system, $S$, is a $2 \times 2$ unitary matrix, and $S_1$ and $S_2$ are just complex numbers of modulus 1; that is, $S_1 = e^{i\phi_1}$ and $S_2 = e^{i\phi_2}$, where $\phi_1$ and $\phi_2$ are known as eigenphases. The optical matrices $\langle S_1 \rangle$ and $\langle S_2 \rangle$ are subunitary complex numbers: for the sake of simplicity, without of generality, we assume that $\langle S_1 \rangle = \langle S_2 \rangle = w$, with $w$ being a real number smaller than one. Under this condition, $\langle r \rangle = \langle r' \rangle = w$ and $\langle t \rangle = 0$, according to Equation (7), such that the optical matrix of the system becomes

$$\langle S \rangle = \begin{pmatrix} w & 0 \\ 0 & w \end{pmatrix} = w \, \mathbb{I}_2 \,. \tag{10}$$

The statistical distribution of $S$ is given by (see Equation (8))

$$dP_w(\phi_1, \phi_2) = \frac{1 - w^2}{\left|1 - e^{i\phi_1}w\right|^2} \frac{1 - w^2}{\left|1 - e^{i\phi_2}w\right|^2} \frac{d\phi_1}{2\pi} \frac{d\phi_2}{2\pi}, \tag{11}$$

which is independent of the symmetry class $\beta$.

Theoretically, we predict that the statistical distribution of the transmission coefficient, in the presence of direct processes, is given by (see Section 4)

$$P_{\theta,w}(T) = \frac{\Theta\left(\sin^2 2\theta - T\right)}{\pi\sqrt{T\left(\sin^2 2\theta - T\right)}} \frac{\left(1 - w^4\right)\sin^2 2\theta}{\left(1 + w^2\right)^2 \sin^2 2\theta - 4w^2\left(\sin^2 2\theta - T\right)}, \tag{12}$$

where $\Theta(x)$ is the Heaviside step function. Taking advantage of this result, the first two moments of the distribution were determined: the ensemble average and variance of $T$ are given by

$$\langle T \rangle_{\theta,w} = \frac{1}{2}\left(1 - w^2\right)\sin^2 2\theta, \tag{13}$$

$$\text{var}(T)_{\theta,w} = \frac{1}{8}\left(1 - w^4\right)\sin^4 2\theta. \tag{14}$$

Two cases are worth mentioning. First, for $\theta = \frac{\pi}{4}$, in the presence of direct processes:

$$P_w(T) = \frac{1}{\pi\sqrt{T(1 - T)}} \frac{\left(1 - w^4\right)}{\left(1 + w^2\right)^2 - 4w^2(1 - T)}. \tag{15}$$

This case was obtained in reference [4] in connection with chaotic cavities with left-right symmetry, in the presence of time reversal invariance and direct processes. The average and variance of $T$ are given by $\langle T \rangle_w = \frac{1}{2}(1 - w^2)$ and $\mathrm{var}(T)_w = \frac{1}{8}(1 - w^4)$, respectively.

Second, the effect of the rotation angle can be observed by turning off the direct processes by setting $w = 0$ in Equation (12); in this case, we obtain

$$P_\theta(T) = \frac{\Theta(\sin^2 2\theta - T)}{\pi \sqrt{T(\sin^2 2\theta - T)}}. \tag{16}$$

For this case, $\langle T \rangle_\theta = \frac{1}{2}\sin^2 2\theta$ and $\mathrm{var}(T)_\theta = \frac{1}{8}\sin^4 2\theta$. The effect of the angle is shown in Figure 1, where we observe that the domain of $T$ is reduced to $[0, \sin^2 2\theta]$. The particular case $\theta = \frac{\pi}{4}$ reproduces the known result for chaotic cavities with left–right symmetry in the absence of direct processes [1] (solid black curve in the left panel of Figure 1).

The combined effect of both $\theta$ and $w$ is shown in the right panel of Figure 1 for $\theta = \frac{3}{5}(\pi/4)$ and $w = 0, 0.5, 0.75$, and $0.9$. While the angle affects the domain of $T$, higher values of $w$ increase the probability of lower values of $T$. What is interesting to note is that all curves match the one that corresponds to $\theta = \frac{\pi}{4}$ by scaling $T$ as $T/\sin^2 2\theta$. If $\tau = T/\sin^2 2\theta$, its statistical distribution is given by

$$p_w(\tau) = \frac{\Theta(1 - \tau)}{\pi \sqrt{\tau(1 - \tau)}} \frac{(1 - w^4)}{(1 + w^2)^2 - 4w^2(1 - \tau)}, \tag{17}$$

which is the same result as in Equation (15).

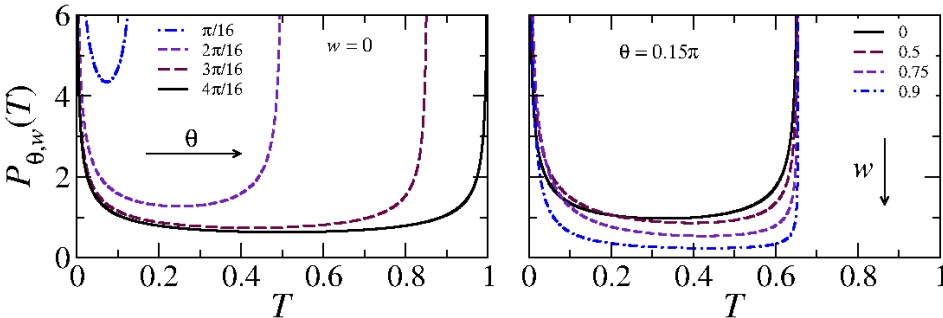

**Figure 1.** (Color online) Dimensionless conductance distribution $P_{\theta,w}(T)$ for $N = 1$. (Left panel) $P_{\theta,w}(T)$ in the absence of direct processes ($w = 0$) for several values of $\theta$. (Right panel) $P_{\theta,w}(T)$ for $\theta = \frac{3}{5}(\pi/4)$, in the presence of direct processes: $w = 0, 0.5, 0.75$, and $0.9$.

### 2.2. Statistical Distribution of T for N = 2 in the Absence of Direct Processes

The scaling that leads to Equation (17) could be predicted directly from the expression of $t$ in Equation (4). This means that the existing results for the distribution of $T$ for $N = 2$ and $\theta = \frac{\pi}{4}$, in the absence of direct processes [1], are valid for an arbitrary value of $\theta$, but replacing $T$ by $\tau$. Therefore, we conjecture that the distribution of $T$ is given by

$$P_\theta^{(1)}(T) = \frac{\Theta(2\sin^2 2\theta - T)}{\pi \sin^2 2\theta} \ln \frac{\sin^2 2\theta + \sqrt{T(2\sin^2 2\theta - T)}}{|\sin^2 2\theta - T|} \tag{18}$$

for $\beta = 1$ and

$$P_\theta^{(2)}(T) = \frac{\Theta(2\sin^2 2\theta - T)}{\pi \sin^2 2\theta} \frac{T}{\sin^4 2\theta} \left( 2\sin^2 2\theta - T \right) F\left( \frac{1}{2}, \frac{3}{2}; 2; \frac{T}{\sin^4 2\theta} \left( 2\sin^2 2\theta - T \right) \right) \tag{19}$$

for $\beta = 2$, where $F(\cdots)$ is the hypergeometric function.

In Figure 2, we compare our conjecture to the results from numerical simulations obtained from the circular ensembles of random-matrix theory (see Appendix A). The excellent agreement between the analytical expressions, Equations (18) and (19), and the numerical simulations for $\beta = 1, 2$, verifies that our conjecture is correct. Additionally, we present the distribution of $T$, obtained from the numerical simulation, for $\beta = 4$, which is compared with the analytical results for $\beta = 1, 2$, in the right panel of Figure 2.

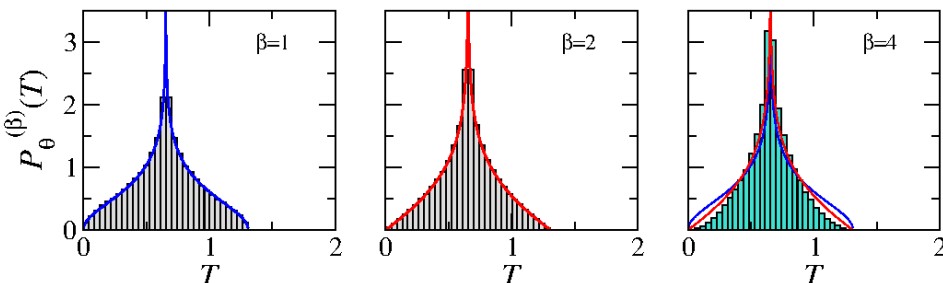

**Figure 2.** (Color online) Dimensionless conductance distribution for $N = 2$ and $\theta = \frac{3}{5}(\pi/4)$, in the absence of direct processes. The histograms corresponds to random-matrix theory simulations for each symmetry class; the continuous lines correspond to Equations (18) and (19) for $\beta = 1$ (left panel) and $\beta = 2$ (middle panel), respectively.

### 2.3. Average and Variance of T in the Absence of Direct Processes for Arbitrary N

The first two moments of the distribution of $T$ for arbitrary $N$, in the absence of direct processes, are easily obtained from the known results for $\theta = \frac{\pi}{4}$.

For the $\beta = 1$ and $\beta = 2$ cases, the transmission coefficient can be written as

$$T = \frac{1}{4}\left[2N - \mathrm{tr}\left(S_1 S_2^\dagger + S_1^\dagger S_2\right)\right]\sin^2 2\theta, \tag{20}$$

where $S_1$ and $S_2$ are $N \times N$ scattering matrices; for $\beta = 4$

$$T = \frac{1}{8}\left[4N - \mathrm{tr}\left(S_1 S_2^\dagger + S_1^\dagger S_2\right)\right]\sin^2 2\theta, \tag{21}$$

where $S_1$ and $S_2$ are $2N \times 2N$ self-dual scattering matrices.

Note that, except for the factor $\sin^2 2\theta$, these expressions are the same as those for $\theta = \frac{\pi}{4}$. Therefore, the existing results for the average and variance of $\tau = T/\sin^2 2\theta$, for $\beta = 1$ and $\beta = 2$, are still valid [1]. For $\beta = 4$, we calculate those quantities in Appendix B. The results for all symmetries are expressed in a single equation for the average and variance of $T$; those are

$$\langle T \rangle_\theta = \frac{N}{2}\sin^2 2\theta \tag{22}$$

and

$$\mathrm{var}(T)_\theta = \frac{N}{4(N\beta + 2 - \beta)}\sin^4 2\theta. \tag{23}$$

### 3. Discussion

The statistical distribution of the scattering matrix of the system, $S$, expressed as the product of two independent Poisson kernels in Equations (8) and (5) for arbitrary $N$, or Equation (11) for $N = 1$, was proposed in reference [4] to describe chaotic scattering by systems with reflection symmetry, in the presence of time-reversal invariance and direct processes. There, also, the direct processes are quantified by an optical matrix of the form of Equation (10). Similarly to that case, this optical matrix can be interpreted as physically realized by two identical tunnel barriers added to our two lead system (diagonalizable by a rotation $\theta$) with no direct processes. This equivalence is not surprising because the

scattering matrix of left–right symmetric systems in the presence of time reversal symmetry are diagonalized by a rotation by an angle $\frac{\pi}{4}$, which is a particular case of a rotation by an angle $\theta$.

The effect of the angle of rotation does appear in the distribution of the transmission coefficient $T$ because it is defined in the original (no rotated) basis. This is shown in Equation (12) for the $N = 1$ case, where the dependence on the angle of rotation, and on the strength $w$ of the direct processes, is clear. Independently of $w$, the domain of $T$ is restricted to $[0, \sin^2 2\theta]$, as can be observed in Equations (12) and (16), and it is valid in the presence and absence of direct processes, respectively. Equation (12) also reduces to Equation (15), which is the known result for left–right symmetric systems in the presence of time reversal symmetry. The effect on the rotation is also observed in the $N = 2$ case, in the absence of direct processes. As can be seen in Equations (18) and (19), for the orthogonal and unitary symmetries, and in Figure 2 for the three symmetry classes, the transmission coefficient domain is restricted to $[0, 2\sin^2 2\theta]$. For arbitrary $N$, it is expected that $T$ is restricted to $[0, N\sin^2 2\theta]$; in fact, according to Equation (22), the average of $T$ scales as $N$.

Since the statistics of the transmission coefficient through disorderless lattices of all sizes, at the band edge, also coincide with the one of the ensembles of chaotic cavities with left–right symmetry in the presence of direct processes [14], it is left for future work to investigate whether systems whose scattering matrix is diagonalizable by a rotation by an angle $\theta$, describe the fluctuations of the transmission coefficient of the disorderless lattices inside the band.

## 4. Method

For $N = 1$, the scattering matrices $S_1$ and $S_2$ are just complex numbers of modulus 1: $S_1 = \mathrm{e}^{\mathrm{i}\phi_1}$ and $S_2 = \mathrm{e}^{\mathrm{i}\phi_2}$. From Equation (4), we see that, for this case, the transmission amplitude is given by

$$t(\phi_1, \phi_2) = \left(\mathrm{e}^{\mathrm{i}\phi_1} - \mathrm{e}^{\mathrm{i}\phi_2}\right)\sin\theta\cos\theta,\tag{24}$$

and the transmission coefficient is defined by $T = |t(\phi_1, \phi_2)|^2$.

By definition, the statistical distribution of $T$ can be determined by

$$P_{\theta,w}(T) = \int \delta\Big[T - |t(\phi_1, \phi_2)|^2\Big]\,\mathrm{d}P_w(\phi_1, \phi_2),\tag{25}$$

where $\delta(x)$ is the delta function. Explicitly, this definition can be expressed as

$$P_{\theta,w}(T) = \frac{(1 - w^2)^2}{4\pi^2}\int_0^{2\pi}\mathrm{d}\phi_1\int_0^{2\pi}\mathrm{d}\phi_2\,\frac{\delta\left\{T - \frac{1}{2}\sin^2 2\theta[1 - \cos(\phi_1 - \phi_2)]\right\}}{[1 + w^2 - 2w\cos\phi_1][1 + w^2 - 2w\cos\phi_2]}.\tag{26}$$

To solve the integral, we define $\psi = \frac{1}{2}(\phi_1 + \phi_2)$ and $\psi' = \frac{1}{2}(\phi_1 - \phi_2)$, such that the limits of the integral change to $\psi' \in [-\psi, \psi]$ for $\psi \in [0, \pi]$ and $\psi' \in [-(2\pi - \psi), 2\pi - \psi]$ for $\psi \in [\pi, 2\pi]$. After some manipulations, the integral is reduced to

$$P_{\theta,w}(T) = \frac{(1 - w^2)^2}{\pi^2}\int_0^\pi\mathrm{d}\psi\int_0^\pi\mathrm{d}\psi'\,\frac{\delta\left(T - \sin^2 2\theta\,\sin^2\psi'\right)}{[1 + w^2 - 2w\cos(\psi + \psi')][1 + w^2 - 2w\cos(\psi - \psi')]}.\tag{27}$$

Looking for the roots of the argument of the delta function in terms of the variable $\psi'$ we find two roots, $\psi_1'$ and $\psi_2'$, defined through the relation $T = \sin^2 2\theta\,\sin^2\psi_j'$ for $j = 1, 2$, such that the integral can be written as

$$P_{\theta,w}(T) = \frac{(1 - w^2)^2}{2\pi^2\sin^2 2\theta}\int_0^\pi\mathrm{d}\psi\int_0^\pi\mathrm{d}\psi'\,\frac{\delta(\psi' - \psi_1') + \delta(\psi' - \psi_2')}{\cos\phi\sin\phi[1 + w^2 - 2w\cos(\psi + \psi')][1 + w^2 - 2w\cos(\psi - \psi')]}.\tag{28}$$

The integration with respect to $\psi'$ leads to

$$P_{\theta,w}(T) = \frac{(1 - w^2)^2}{c\pi^2}\frac{\Theta\left(\sin^2 2\theta - T\right)}{\sqrt{T\left(\sin^2 2\theta - T\right)}}\int_0^\pi\frac{1}{a + b\cos\psi + \cos^2\psi}\,\mathrm{d}\psi,\tag{29}$$

where

$$a = \frac{1}{c}\left[(1+w^2)^2 - 4w^2 \frac{T}{\sin^2 2\theta}\right], \quad b = \frac{4}{c}w(1+w^2)\sqrt{\frac{\sin^2 2\theta - T}{\sin^2 2\theta}}, \quad c = 4w^2, \quad (30)$$

and $\Theta(x)$ is the Heaviside step function. The remaining integration was performed in reference [4]; using such result, we finally arrive to the expression of Equation (12).

Once $P_{\theta,w}(T)$ has been obtained, it is interesting to calculate its first two moments and the variance of $T$. The $n$th moment is defined by

$$\langle T^n \rangle_{\theta,w} = \int_0^1 T^n P_{\theta,w}(T)\, dT. \quad (31)$$

Explicitly, the first moment is given by

$$\langle T \rangle_{\theta,w} = \int_0^{\sin^2 2\theta} \frac{1}{\pi\sqrt{\sin^2 2\theta - T}} \frac{\sqrt{T}(1-w^4)\sin^2 2\theta}{(1+w^2)^2 \sin^2 2\theta - 4w^2(\sin^2 2\theta - T)}\, dT. \quad (32)$$

If we make $T = \sin^2 2\theta \cos^2 z$, the integral can be transformed to

$$\langle T \rangle_{\theta,w} = \frac{2(1-w^2)\sin^2 2\theta}{\pi(1+w^2)} \int_0^{\pi/2} \frac{\cos^2 z}{\sigma^2 \sin^2 z - 1}\, dz, \quad \text{with} \quad \sigma = \frac{2w}{1+w^2}. \quad (33)$$

Using trigonometric identities which relate $\cos z$ and $\sin z$ with $\sec z$ and $\tan z$, the last integral can be written as

$$\int_0^{\pi/2} \frac{\cos^2 z}{\sigma^2 \sin^2 z - 1}\, dz = \int_0^{\pi/2} \frac{\sec^2 z}{(1+\tan^2 z)\left[(\sigma^2 - 1)\tan^2 z - 1\right]}\, dz. \quad (34)$$

Now, changing to the variable $v = \tan z$, we obtain that

$$\int_0^{\pi/2} \frac{\cos^2 z}{\sigma^2 \sin^2 z - 1}\, dz = \int_0^\infty \frac{1}{(1+v^2)[(\sigma^2 - 1)v^2 - 1]}\, dv$$

$$= \frac{1}{\sigma^2} \int_0^\infty \frac{1}{1+v^2}\, dv - \frac{1-\sigma^2}{\sigma^2} \int_0^\infty \frac{1}{1 + (1-\sigma^2)v^2}\, dv. \quad (35)$$

The integral in the first term of the right hind side is well known, whereas for the integral in the second term, we change to the variable $u = \sqrt{1-\sigma^2}v$; the sum of both terms gives

$$\int_0^{\pi/2} \frac{\cos^2 z\, dz}{\sigma^2 \sin^2 z - 1} = \frac{\pi}{2\sigma^2}\left(1 - \sqrt{1-\sigma^2}\right). \quad (36)$$

Therefore, by substituting this result into Equation (33) we finally arrive at the result expressed in Equation (13).

Similarly, following the same procedure, the expression for the second moment can be written as

$$\langle T^2 \rangle_{\theta,w} = \frac{2(1-w^2)\sin^4 2\theta}{\pi(1+w^2)\sigma^2}\left[\frac{1-\sigma^2}{\sigma^2}\int_0^\infty \frac{dv}{1+v^2} - \frac{(1-\sigma^2)^2}{\sigma^2}\int_0^\infty \frac{dv}{1+(1-\sigma^2)v^2} - \int_0^\infty \frac{dv}{(1+v^2)^2}\right]. \quad (37)$$

The integrals in the first two terms on the right-hand side were previously solved, and the integral in the last term is easily solved by the change of variables $v = \sin y$. Then,

$$\langle T^2 \rangle_{\theta,w} = \frac{1}{8}\left(1 - w^2\right)\left(3 - w^2\right)\sin^4 2\theta. \quad (38)$$

Using the results for the first and second moments, we can determine the variance of $T$ through the definition $\mathrm{var}(T)_{\theta,w} = \langle T^2 \rangle_{\theta,w} - \langle T \rangle_{\theta,w}^2$. The result is shown in Equation (14).

## 5. Conclusions

We studied the statistics of the scattering and transport properties through systems whose associated scattering matrices are diagonalizable, by a rotation by an arbitrary but fixed angle, into two independent scattering matrices. Similarly to what happens for left–right symmetric chaotic cavities in the presence of direct processes, the statistical distribution of the scattering matrix of the system is given by the product of two independent Poisson kernels. As a consequence, the statistical distribution of the dimensionless conductance is affected by the rotation angle and the direct processes intensity, but, in the absence of direct processes, it reduces to well known results for left–right symmetric systems with time reversal invariance. We expect that our investigation may help in the understanding the fluctuations of the dimensionless conductance inside the band of disorderless lattices of all sizes, as happened at the band edge.

**Author Contributions:** Conceptualization, F.C.-R. and M.M.-M.; formal analysis, F.C.-R. and M.M.-M.; investigation, F.C.-R. and M.M.-M.; methodology, F.C.-R. and M.M.-M.; project administration, M.M.-M.; Supervision, M.M.-M.; visualization, M.M.-M.; writing—original draft, F.C.-R.; writing—review and editing, M.M.-M. All authors have read and agreed to the published version of the manuscript.

**Funding:** This research was funded by Consejo Nacional de Ciencia y Tecnología (CONACyT) grant number CB-2016/285776.

**Data Availability Statement:** Not applicable.

**Acknowledgments:** Felipe Castañeda-Ramírez also akcnowledges financial support from CONACyT.

**Conflicts of Interest:** The authors declare no conflict of interest.

## Appendix A. Parameterization of Circular Ensembles for $N = 2$

A useful parameterization of a unitary scattering matrix $S'$ is the polar parameterization [22]

$$S' = \begin{pmatrix} U & 0 \\ 0 & V \end{pmatrix} \begin{pmatrix} -\sqrt{1-\tau} & \tau \\ \tau & \sqrt{1-\tau} \end{pmatrix} \begin{pmatrix} U' & 0 \\ 0 & V' \end{pmatrix}, \tag{A1}$$

where $\tau$ is a diagonal matrix whose diagonal elements are $\tau_1, \ldots, \tau_N$, for $\beta = 1, 2$, and $\mathbb{I}_2 \tau_1, \ldots, \mathbb{I}_2 \tau_N$ for $\beta = 4$, with $0 \leq \tau_1, \tau_2, \ldots, \tau_N < 1$. Here, $U$, $V$, $U'$, and $V'$ are arbitrary unitary matrices of rank $N$ for $\beta = 2$, $U' = U^T$ and $V' = V^T$ for $\beta = 1$. For $\beta = 4$, $U' = U^R$ and $V' = V^R$, where $U$ and $V$ are unitary matrices of rank $2N$ and [23]

$$U^R = -Z U^T Z, \tag{A2}$$

being $U^T$ the transpose of $U$ and $Z$ is a block diagonal matrix with all the diagonal elements equal to

$$Z_2 = \begin{pmatrix} 0 & 1 \\ -1 & 0 \end{pmatrix}. \tag{A3}$$

For the particular case $N = 2$, $U = \mathrm{e}^{\mathrm{i}\alpha}$, $V = \mathrm{e}^{\mathrm{i}\gamma}$, $U' = \mathrm{e}^{\mathrm{i}\alpha'}$, and $V' = \mathrm{e}^{\mathrm{i}\gamma'}$ for $\beta = 2$, where $0 \leq \alpha, \alpha', \gamma, \gamma' < 2\pi$; $\alpha' = \alpha$ and $\gamma' = \gamma$ for $\beta = 1, 4$. For $\beta = 4$, it is convenient to use the Hurwitz parameterization of a $2 \times 2$ unitary matrix [23], namely,

$$U = \mathrm{e}^{\mathrm{i}\alpha} E \quad \text{with} \quad E = \begin{pmatrix} a & b \\ -b^* & a^* \end{pmatrix}, \tag{A4}$$

where

$$a = \mathrm{e}^{\mathrm{i}\psi} \cos\phi \quad \text{and} \quad b = \mathrm{e}^{\mathrm{i}\chi} \cos\phi, \tag{A5}$$

with $0 \leq \phi < \frac{\pi}{2}$ and $0 \leq \alpha, \psi, \chi < 2\pi$. Therefore, Equation (A1) is written as

$$S' = \begin{bmatrix} -\sqrt{1-\tau}\,\mathrm{e}^{\mathrm{i}(\alpha+\alpha')}J & \sqrt{\tau}\,\mathrm{e}^{\mathrm{i}(\alpha+\gamma')}D \\ \sqrt{\tau}\,\mathrm{e}^{\mathrm{i}(\alpha'+\gamma)}D^\dagger & \sqrt{1-\tau}\,\mathrm{e}^{\mathrm{i}(\gamma+\gamma')}J \end{bmatrix}, \tag{A6}$$

where, $J = D = 1$ for $\beta = 1, 2$; $J = \mathbb{I}_2$ and $D = EE'^{\dagger}$ for $\beta = 4$.

A circular ensemble is defined through the invariant measure $d\mu_{\beta}(S')$, which expresses the equal a priori distribution for $S'$ [19]. The corresponding invariant measure for $S'$ of Equation (A6) is given by [19,22]

$$d\mu_{\beta}(S') = \frac{\beta}{2}\tau^{\beta/2-1}\,d\tau\,\frac{d\alpha}{2\pi}\frac{d\gamma}{2\pi} \times \begin{cases} 1 & \text{for} \quad \beta = 1, \\ \dfrac{d\alpha'}{2\pi}\dfrac{d\gamma'}{2\pi} & \text{for} \quad \beta = 2, \\ d\mu(E)\,d\mu(E') & \text{for} \quad \beta = 4, \end{cases} \tag{A7}$$

where $d\mu(E)$ and $d\mu(E')$ are of the form

$$d\mu(E) = \sin(2\phi)\,d\phi\,\frac{d\psi}{2\pi}\frac{d\chi}{2\pi}. \tag{A8}$$

Numerical simulations of the circular ensembles can be implemented by generating random numbers according to Equation (A7) for the scattering matrix $S'$ of Equation (A6). In this sense, scattering matrix of Equation (3), is generated by two independent realizations of $S'$, that represent $S_1$ and $S_2$ in Equations (3) and (4).

### Appendix B. Calculation of the Average and Variance of $T$ for $\beta = 4$, for Arbitrary $N$

For the symplectic symmetry the transmission coefficient is given by

$$T = \frac{1}{2}\mathrm{tr}\left(tt^{\dagger}\right), \tag{A9}$$

where the factor one-half in front is due to Kramer degeneracy and $S_1$ and $S_2$ are $2N \times 2N$ self-dual matrices. Defining $\tau$ as $\tau = T/\sin^2 2\theta$, we have that

$$\tau = \frac{1}{8}\left[4N - \mathrm{tr}\left(S_1 S_2^{\dagger} + S_1^{\dagger}S_2\right)\right] = \frac{N}{2} - \frac{1}{8}\sum_{i=1}^{2N}\left[\left(S_1 S_2^{\dagger}\right)_{ii} + \left(S_1^{\dagger}S_2\right)_{ii}\right]. \tag{A10}$$

To calculate the average and variance of $\tau$, it is convenient to parameterize $S_1$ and $S_2$ as $S_1 = UU^R$ and $S_2 = VV^R$, where $U$ and $V$ are unitary matrices of rank $2N$, that satisfy Equation (A2). In this way, the average and variance of $\tau$ are reduced to averages over the unitary group. The average of $\tau$ is given by

$$\langle \tau \rangle = \frac{N}{2} + \frac{1}{4}\sum_{i,j,k,l,m,n=1}^{2N} Z_{jk}Z_{mn}Q_{ij,kl}Q^{lm,in} = \frac{N}{2}, \tag{A11}$$

where [24]

$$Q^{a_1\alpha_1,\ldots,a_l\alpha_l}_{b_1\beta_1,\ldots,b_m\beta_m} = \left\langle \left(U_{b_1\beta_1}\cdots U_{b_m\beta_m}\right)\left(U_{a_1\alpha_1}\cdots U_{a_l\alpha_l}\right)^{*}\right\rangle, \tag{A12}$$

where $\langle\cdots\rangle$ denotes the average on the unitary group. In the last equality of Equation (A11) we have used that $Q^{a_1\alpha_1,\ldots,a_l\alpha_l}_{b_1\beta_1,\ldots,b_m\beta_m}$ is zero for $m \neq l$ [24]. Similarly, the second moment of the distribution of $\tau$ can be written as

$$\left\langle \tau^2 \right\rangle = \frac{N^2}{4} + \frac{1}{32}\sum_{\substack{i',j',k',l',m',n', \\ i,j,k,l,m,n=1}}^{2N} Z_{jk}Z_{mn}Z_{j'k'}Z_{m'n'}Q^{l'm',i'n'}_{ij,lk}Q^{lm,in}_{i'j',l'k'}. \tag{A13}$$

Here, we used the result [24]

$$Q^{a\alpha,b\beta}_{a'\alpha',b'\beta'} = \frac{1}{N^2-1}\left(\Delta^{ab}_{a'b'}\Delta^{\alpha\beta}_{\alpha'\beta'} + \Delta^{ba}_{a'b'}\Delta^{\beta\alpha}_{\alpha'\beta'}\right) - \frac{1}{N(N^2-1)}\left(\Delta^{ab}_{a'b'}\Delta^{\beta\alpha}_{\alpha'\beta'} + \Delta^{ba}_{a'b'}\Delta^{\alpha\beta}_{\alpha'\beta'}\right), \tag{A14}$$

where $\Delta_{a'b'}^{a\,b} = \delta_{a'}^{a}\delta_{b'}^{b}$ and $\delta_{a'}^{a}$ is the Kronecker delta, to find

$$\left\langle \tau^2 \right\rangle = \frac{N^2}{4} + \frac{N}{4(4N-2)} \tag{A15}$$

and

$$\text{var}(\tau) = \frac{N}{4(4N-2)}, \tag{A16}$$

which lead to the results expressed in Equations (22) and (23).

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
