# Peer review of "Blocky Diagonalized Scattering Matrices in Chaotic Scattering with Direct Processes"

_quantumrep, doi:10.3390/quantum5010002_

Round 1

Reviewer 1 Report

While the study of transmission/ scattering properties of S-matrices of the particular class of systems considered seems interesting, in my opinion the following issues should be addressed:

1. I think the introduction needs to be elaborated on a bit more. In particular, it is not clear what the motivation for studying the N=1 case is if the generic case when the rank of the matrix is 2N is known.

2. I couldn't find a definition of  direct processes and a justification of why it is related to w anywhere in the text. Although the relevant literature has been cited, it is good to add a brief explanation for this.

3. I think it is better to merge Sections 3 and 5 together to provide an overall summary of the paper by referencing the relevant sections.

Reviewer 2 Report

This paper addresses complex (chaotic) scattering problems in systems displaying general symmetries, not restricted to mirror reflections. The paper is in general well-written and I think it is free from obvious errors, though in my opinion, and admitting that I am not a native English speaker, I think it requires moderate language revisions.

While I think the results presented by the authors are interesting and may merit publication, I think the present manuscript is not yet ready for that. In particular, I felt somewhat disappointed from the calculations and results actually presented, and what I think is promised in the abstract: The abstract is formulated for 2x2 blocks of N-dimensional matrices, while the actual computations only consider the case N=1; the same occurs in the conclusions, which are formulated in more general terms than the N=1 case. I agree that the most general N case might be too difficult to obtain concrete and useful results, but the case N=2 seems to me achievable and actually important for 2d chaotic cavities scattering. This is my main major criticism on the paper, and I urge the authors to include the results for N=2.

Other more concrete points the authors should consider:

- In page 2, it is stated that S_1 and S_2 are *statistically independent* scattering matrices. The authors should provide insights into this, or otherwise state that it is an assumption.

- The curves in Fig 1, left panel, which illustrate the results for N=1 without direct processes (Eq. 16), seem to have the same shape for different values of \theta. That is, there seems to be scaling involving T and P_{\theta,w=0}(T) such that all curves collapse into one, independently from the actual value of \theta; see Eq (16). Presenting the results in that way would be interesting because, then, for N=1 all results under proper scaling would match the case \pi/4.

- Along the same lines of the last item, is there a scaling that for N=2 makes all results collapse?

- Proof read by a native English speaker is highly recomended.

Round 2

Reviewer 2 Report

The authors followed my recommendations. I recommend the paper for publication.